# Dysregulations of Key Regulators of Angiogenesis and Inflammation in Abdominal Aortic Aneurysm

**DOI:** 10.3390/ijms241512087

**Published:** 2023-07-28

**Authors:** Daniel Zalewski, Paulina Chmiel, Przemysław Kołodziej, Grzegorz Borowski, Marcin Feldo, Janusz Kocki, Anna Bogucka-Kocka

**Affiliations:** 1Chair and Department of Biology and Genetics, Medical University of Lublin, 4a Chodźki St., 20-093 Lublin, Poland; pachmiel13@gmail.com (P.C.); przemyslaw.kolodziej@umlub.pl (P.K.); anna.kocka@umlub.pl (A.B.-K.); 2Chair and Department of Vascular Surgery and Angiology, Medical University of Lublin, 11 Staszica St., 20-081 Lublin, Poland; gwborowski@gmail.com (G.B.); martinf@interia.pl (M.F.); 3Department of Clinical Genetics, Chair of Medical Genetics, Medical University of Lublin, 11 Radziwiłłowska St., 20-080 Lublin, Poland; janusz.kocki@umlub.pl

**Keywords:** angiogenesis, inflammation, gene expression, abdominal aortic aneurysm

## Abstract

Abdominal aortic aneurysm (AAA) is a chronic vascular disease caused by localized weakening and broadening of the abdominal aorta. AAA is a clearly underdiagnosed disease and is burdened with a high mortality rate (65–85%) from AAA rupture. Studies indicate that abnormal regulation of angiogenesis and inflammation contributes to progression and onset of this disease; however, dysregulations in the molecular pathways associated with this disease are not yet fully explained. Therefore, in our study, we aimed to identify dysregulations in the key regulators of angiogenesis and inflammation in patients with AAA in peripheral blood mononuclear cells (using qPCR) and plasma samples (using ELISA). Expression levels of *ANGPT1*, *CXCL8*, *PDGFA*, *TGFB1*, *VEGFB*, and *VEGFC* and plasma levels of TGF-alpha, TGF-beta 1, VEGF-A, and VEGF-C were found to be significantly altered in the AAA group compared to the control subjects without AAA. Associations between analyzed factors and risk factors or biochemical parameters were also explored. Any of the analyzed factors was associated with the size of the aneurysm. The presented study identified dysregulations in key angiogenesis- and inflammation-related factors potentially involved in AAA formation, giving new insight into the molecular pathways involved in the development of this disease and providing candidates for biomarkers that could serve as diagnostic or therapeutic targets.

## 1. Introduction

Abdominal aortic aneurysm (AAA) is an underestimated health problem worldwide. The disease is caused by morphological changes in the abdominal aorta, observed mainly as focal weakness and a broadened diameter of the aorta. Due to the asymptomatic character of the disease, AAA is often detected accidentally during imaging tests such as ultrasonography, computed tomography, and, less often, magnetic resonance [1,2,3]. Progressive enlargement of AAA could lead to aneurysm rupture, a complication burdened by a high mortality rate that reaches 85% [4,5]. Due to these reasons, AAA screening programs were conducted in various countries (e.g., Denmark, Sweden, and the United Kingdom). Screening studies showed that the incidence rate of AAA is approximately 1–2% in men over 65 years of age. In women, the incidence is generally lower, but with a worse outcome and prognosis [1,2,6,7,8,9].

AAA is a multifactorial disease with various genetic and environmental risk factors, including older age, male sex, smoking, and positive family history. Studies have shown that the phenotypic variance of AAA is determined by genetics in approximately 70–80%, while the remining part is associated with environmental factors [1,2]. Therefore, ultrasound screening examinations for AAA are recommended in at-risk populations, especially in men and women 65 to 75 years of age with a history of smoking [10].

The molecular mechanisms that contribute to AAA development are associated with dysregulations of such biological processes as ECM (extracellular matrix) degradation, inflammation, angiogenesis, vascular smooth muscle cells apoptosis, endothelial function, and response to oxidative stress [11].

The degradation of ECM mediated by matrix metalloproteases (MMP) and the increase in inflammation emerged as the main pathophysiological mechanisms responsible for the progression, enlargement, and rupture of the aneurysm [10,11]. MMP, together with cysteine proteases of the cathepsin family secreted by macrophages and smooth muscle cells, are responsible for the proteolytic degradation of elastin and collagen in the aortic wall. Some products of ECM cleavage are biologically active and promote cell activation and monocyte/macrophage infiltration. Damaged aortic wall enhances volume of perivascular adipose tissue, which stimulates the expression of inflammatory factors such as resistin, leptin, cytokines, and chemokines, causing further infiltration and activation of neutrophils, macrophages, lymphocytes, and other inflammatory cells at the AAA site. Furthermore, inflammation increases the MMP activity, further stimulating inflammation and remodeling of the aortic wall by altering aortic structural integrity [12,13,14,15].

The weakness of the aneurysmal aortic wall is also excavated by the reduction of the number of vascular smooth muscle cells (VSMC), caused by increased apoptosis [1,12]. Furthermore, signs of enhanced angiogenesis were also observed in AAA, especially within the aortic media of the aneurysm. It is suggested that increased neovascularization in AAA is driven by VEGF signaling, as an increased amount of VEGF positive cells (i.e., macrophages and VSMC) was observed in the aortic wall of AAA [2,16,17,18,19,20].

To date, numerous attempts have been made to identify molecular markers associated with AAA, which could be used as diagnostic or treatment targets. Dysregulations of many proteins (e.g., cathepsin S, high sensitive-CRP, and interleukins), mRNA (e.g., *GGT1*, MMPs), lncRNA (e.g., H19, PVT1, and GAS5), and miRNAs (e.g., miR-21, miR-29b, and miR-155), as well as genetic loci, were previously associated with AAA [21,22,23,24,25,26,27,28,29,30]; however, none of the proposed biomarkers has been introduced into clinical practice so far. Therefore, there is still an urgent need to identify robust molecular biomarkers of AAA and assess their diagnostic value, to select those useful for AAA management. Especially, the most promising could be circulatory biomarkers, which could be detected in highly accessible biological material such as blood or its compartments (e.g., plasma, circulatory cells). Therefore, in the presented study, we investigated dysregulations in the key regulators of angiogenesis and inflammation in the group of AAA patients compared to non-AAA subjects, both in gene expression and plasma protein levels. The aim of our study was to identify potential biomarkers of AAA and to gain new insight into the molecular mechanisms involved in the development of AAA.

## 2. Results

### 2.1. Characterization of the Study Subjects

The study population included two groups of subjects: 40 patients with AAA (AAA group) and 24 healthy volunteers (control group). Demographic and clinical characteristics of these groups are presented in Table 1. No statistically significant differences were found between the groups in age, body mass index (BMI), and plasma levels of HDL, creatinine, urea, and homocysteine, as well as in the occurrence of hypertension, hypercholesterolemia, and hypertriglyceridemia. However, the studied groups differed in some other characteristics, which illustrate risk factors and biochemical alterations associated with AAA. In particular, the AAA group includes more men (85%) than women (15%), while the control group is more balanced. Furthermore, 37.5% of the AAA group constituted smoking subjects, while the control group had none. The AAA subjects also had the highest mean levels of LDL, cholesterol, and C-reactive protein in blood than the control group. In turn, the control group was characterized by a higher mean of fibrinogen levels. Due to the observed differences between the studied groups, potential associations between these characteristics and the expression of the studied genes were investigated, and obtained results are provided in Section 2.4.

### 2.2. Genes Related to Angiogenesis and Inflammation Are Dysregulated in AAA

The expression levels of 10 angiogenesis-associated genes (*ANGPT1*, *ANGPT2*, *FGF2*, *PDGFA*, *PDGFB*, *TGFA*, *TGFB1*, *VEGFA*, *VEGFB*, and *VEGFC*) and 8 inflammation-related genes (*CCL2*, *CCL5*, *CSF2*, *CXCL8*, *IL1A*, *IL1B*, *IL6*, and *TNF*) were analyzed in peripheral blood mononuclear cells (PBMC) of 40 patients with AAA (AAA group) and 24 healthy controls (control group) using the RT-qPCR method. The quality control of the expression data showed good homogeneity between samples and no outliers were detected (Appendix A); therefore, all samples were used for the analysis. Four out of 18 analyzed genes (*ANGPT2*, *CSF2*, *IL1A*, and *IL6*) were excluded from the study due to receiving low signal or unreliable data in more than half of subjects in at least one of the study groups (see the Methods section for more details) (Appendix A). Expression levels of 14 retained genes were compared between AAA and control groups using the ddCt method for relative quantification.

The distributions of normalized expression levels (dCt values) of 14 analyzed genes in the studied samples are presented in Figure 1A. Changes in the expression of these genes between the compared groups are represented as RQ values (Figure 1B, Appendix A). Seven out of the analyzed genes (*ANGPT1*, *CCL2*, *CCL5*, *CXCL8*, *IL1B*, *PDGFA*, and *VEGFC*) showed higher expression in the AAA group compared to the control group, while the expression of the other seven genes (*FGF2*, *PDGFB*, *TGFA*, *TGFB1*, *TNF*, *VEGFA*, and *VEGFB*) was lower in this comparison (Figure 1B). When statistical significance was tested, 6 out of 14 analyzed genes (*ANGPT1*, *CXCL8*, *PDGFA*, *TGFB1*, *VEGFB*, and *VEGFC*) were significantly differentially expressed (*p* < 0.05) between the compared groups (Figure 1B, Table 2).

The performance of classifying subjects into the appropriate group by the expression of six selected genes was further assessed by generating ROC curves (Appendix A). The obtained aeras under the ROC curves ranged from 0.809 to 0.645, showing good classification accuracy (Table 2).

The last method used to evaluate the differential character of six selected genes was logistic regression, run in univariate and multivariate modes. This method was used to calculate odds ratios (ORs) with *p* values, which illustrate the chance for the AAA condition to occur when the dCt values decrease by one unit (the expression increases twice). Univariate logistic regression shows that all six genes had statistically significant OR values (Table 2 and Appendix A, Appendix A). In multivariate logistic regression, sex, age, BMI, and smoking were used as additional explanatory variables to evaluate whether the altered expression of six selected genes depends on these four characteristics. If the gene remains statistically significant when adjusted for these characteristics, its difference between the compared groups was considered independent of the used characteristics. The differential expression of only *VEGFB* was found to be independent in the applied multivariate approach (Appendix A, Appendix A). These results indicate that the expression of selected genes could be associated with other variables; therefore, analysis of such associations was performed and results were provided in the next paragraph.

Performed analyses indicate that the differences in expression of six selected genes have a good ability to distinguish compared groups and could be used to understand the regulation of angiogenesis and inflammation pathways during the AAA development.

### 2.3. Angiogenesis-Related Proteins Are Dysregulated in AAA

Plasma levels of six angiogenesis-related proteins (ANGPT-1, ANGPT-2, TGF-alpha, TGF-beta 1, VEGF-A, and VEGF-C) were analyzed in 40 patients with AAA (AAA group) and 20 healthy controls (control group) using the ELISA method. In samples in which the measurements were out of the range of used ELISA kits, the samples were diluted to fall within the standard curve. The amounts of samples in which the concentrations of the analyzed proteins were detected or not are presented in Appendix A. The PCA analysis showed good homogeneity of the data (Appendix A); therefore, none of the samples were excluded from further analysis. The distributions of plasma levels of six analyzed proteins in the studied groups are presented in Figure 2. The mean concentrations of ANGPT-1 and ANGPT-2 were not statistically significantly different between the AAA and control groups. Plasma levels of TGF-alpha and TGF-beta 1 were not detected in the control group, but were revealed in AAA patients. In the case of proteins of the VEGF family, the mean concentration of VEGF-A was higher in the AAA group than in the control group, while the difference in plasma levels of VEGF-C was the opposite (Table 3, Figure 2). In the ROC analysis, the highest values of areas under ROC curves were obtained for TGF-alpha (AUC-ROC = 1.000), VEGF-C (AUC-ROC = 0.931), and TGF-beta 1 (AUC-ROC = 0.900).

### 2.4. Relationships with Risk Factors and Biochemical Parameters

The differential character of six selected genes and analyzed proteins could be not only the results of the disease presence, but could also be associated with risk factors and biochemical abnormalities related to these diseases. Potential associations between the expression of six selected genes or plasma levels of analyzed proteins and continuous-type characteristics (age, BMI, and blood levels of cholesterol, LDL, HDL, homocysteine, urea, fibrinogen, creatinine, and C-reactive protein) were explored using correlation and univariate linear regression analyses. For genes, only weak correlations were obtained (R < 0.35) and those with statistical significance have correlation coefficients between 0.35 and 0.27 (Appendix A). In linear regression analysis, only three associations were confirmed as statistically significant: between fibrinogen blood levels and expression of *TGFB1* and *VEGFB*, as well as between homocysteine blood levels and expression of *VEGFB* (Table 4). In relation to analysis of proteins, higher correlation coefficients were obtained. Nine of them were statistically significant (Appendix A), including seven correlations confirmed by simple linear regression (Table 4).

The evaluation of the relationships was also performed for categorical-type characteristics (sex, smoking, hypertension, hypercholesterolemia, and hypertriglyceridemia) using the Mann–Whitney U-test and the Student’s *t*-test, depending on the normality of the data. Statistically significant differences were found for the expression of *PDGFA* (*p* = 0.0030), *VEGFB* (*p* = 0.0055), and *ANGPT1* (*p* = 0.0088) between smokers and nonsmokers, as well as for the expression of *VEGFB* (*p* = 0.0054) between men and women (Appendix A). Regrading proteins, plasma levels of ANGPT-1 (*p* = 0.0304), TGF-alpha (*p* = 0.0027), and VEGF-C (*p* = 0.0226) were statistically significantly associated with smoking, while plasma levels of TGF-alpha were associated with sex (*p* = 0.0207) (Appendix A).

Obtained results showed that among the genes, the highest number of associations were found for *VEGFB* (with fibrinogen, homocysteine, smoking and sex variables), while among proteins, there was TGF-alpha (associated with CRP, cholesterol, smoking, and sex variables).

### 2.5. Coexpression of Selected Genes and Proteins

Genes and proteins with changed expression in the same direction could exert a similar expression pattern in the studied subjects. To investigate this similarity, pairwise correlation analysis was performed for six genes selected as significantly differentially expressed in the AAA vs. control group and the analyzed proteins (Figure 3, Appendix A). Strong positive correlations were found between expression levels of *ANGPT1*, *PDGFA*, and *VEGFC*, whose correlations coefficients were above 0.8 (Appendix A). Moderately positive correlations were found between *TGFB1* and *VEGFB* expression levels (R = 0.67, *p* = 4.68 × 10^−9^), as well as between *ANGPT1* and *CXCL8* expression levels (R = 0.6, *p* = 4.51 × 10^−7^). Observed correlations between genes may indicate their coexpression, functional association, and common regulatory mechanisms occurring in AAA.

Regarding protein–protein correlations, the strongest correlation was found between plasma levels of VEGF-C and TGF-alpha (R = −0.55, *p* = 6.69 × 10^−6^), while among gene–protein correlations, the strongest was between *TGFB1* and ANGPT-1 levels (R = −0.52, *p* = 1.79 × 10^−5^).

Especially interesting correlations were found between the expression of genes and plasma levels of their encoding proteins. The only statistically significant was the weak correlation between expression of *ANGPT1* and plasma levels of ANGPT-1 (R = 0.35, *p* = 0.006). It could be concluded that the expression of *ANGPT1*, *TGFA*, *TGFB1*, *VEGFA*, and *VEGFC* in PBMC is probably not reflected by protein concentration in the plasma compartment.

### 2.6. The Expression of the Analyzed Genes Are Unable to Predict the Diameter of the Aneurysm in the Studied Patients

To assess whether the expression of 14 analyzed genes and 6 proteins could be indicative of the aneurysm diameter, we investigated the relationships between the normalized expression of the analyzed genes and the measurements of aneurysm diameter in the studied patients with AAA. Associations were explored using correlation and univariate linear regression analysis. There was one statistically significant correlation between aneurysm diameter and plasma levels of TGF-beta 1 (R = 0.32, *p* = 0.045), but in the regression analysis this association was not statistically significant. To conclude, the expression of the analyzed genes and proteins is not significantly associated with the aneurysm diameter in the studied patients with AAA.

## 3. Discussion

In the presented study, differences in the expression levels of 10 key regulators of angiogenesis (*ANGPT1*, *ANGPT2*, *FGF2*, *PDGFA*, *PDGFB*, *TGFA*, *TGFB1*, *VEGFA*, *VEGFB*, and *VEGFC*) and 8 main regulators of inflammation (*CCL2*, *CCL5*, *CSF2*, *CXCL8*, *IL1A*, *IL1B*, *IL6*, and *TNF*), as well as in plasma levels of 6 proteins (ANGPT-1, ANGPT-2, TGF-alpha, TGF-beta 1, VEGF-A, and VEGF-C) were analyzed between group of patients with AAA and group of volunteers without AAA. Higher expression of *ANGPT1*, *CXCL8*, *PDGFA*, and *VEGFC*, as well as higher plasma levels of TGF-alpha, TGF-beta 1, and VEGF-A, were found to be statistically significant. In turn, lower expression of *TGFB1* and *VEGFB*, as well as lower plasma levels of VEGF-C, were found to be statistically significant (Table 2 and Table 3, Figure 1 and Figure 2). These results suggest that dysregulations in the main regulators of angiogenesis and inflammation could be involved in the etiopathogenesis of AAA. This conclusion is in agreement with the results of other studies on this topic.

The angiopoietins investigated in this study had a different expression pattern in the PBMC samples of the studied subjects. *ANGPT1* expression was detected in most samples, while *ANGPT2* was not detected in almost all samples (Appendix A). In plasma, the levels of both angiopoietins seem to be similar (Figure 2).

In this study, a higher expression of *ANGPT1* was found in the AAA group compared to the control group (Table 2, Figure 1). ANGPT-1 regulates endothelium maintenance and blood vessel growth through interactions with Tie-2 and integrin receptors. This protein is necessary for proper angiogenesis, because its knockout in animal studies caused defects in heart development and a reduction in vascular complexity. A previous study showed that in AAA, ANGPT-1 upregulates the expression of apelin, which has been shown to reduce aortic diameter and inhibit elastin and collagen degradation. Furthermore, ANGPT-1 inhibits inflammatory cell infiltration and promotes neovascularization by increasing the proliferation of endothelial cells [31]. These results suggest a potential protective effect of this protein against aneurysm. This conclusion was confirmed in another study, where endothelial progenitor cells with overexpressed ANGPT-1 were proposed as a promising strategy for the treatment of aneurysm due to their strong ability to improve organization of fibrotic components in the aneurysms and enhance angiogenesis [32]. Furthermore, in patients with aneurysmal subarachnoid hemorrhage, higher serum levels of ANGPT-1 at 72 h after hemorrhage were associated with a good outcome [33]. To conclude, upregulation of *ANGPT1* shown in the presented study could indicate enhanced angiogenesis in patients with AAA; however, more studies are needed to confirm these conclusions.

The gene with the highest difference in expression between the AAA and control groups was *CXCL8* (Table 2, Figure 1), which encodes interleukin-8 (IL-8). This result is consistent with previous studies, in which higher levels of IL-8 RNA and protein were demonstrated in AAA tissues compared to normal tissues [34,35]. A higher expression of IL-8 was also associated with mural inflammation and increased infiltration of Th lymphocytes into the AAA wall [35]. In animal models of AAA, inhibition of CXCR1/2 receptors for IL-8 caused reduced aneurysm growth; therefore, targeting the signaling pathway of this cytokine could be a promising strategy for AAA therapy [34]. Higher levels of IL-8 in cerebrospinal fluid and blood were also associated with intracranial aneurysm and were related to the aneurysm size [36,37]. Our study confirms that enhanced IL-8 signaling could be involved in aneurysm development and could serve as a potential diagnostic and therapeutic target.

Among the two analyzed genes belonging to the PDGF family, *PDGFA* was demonstrated to have higher expression in AAA versus control groups in comparison with statistical significance (Table 2, Figure 1), while *PDGFB* was not significantly downregulated. This result suggests that enhanced angiogenesis is involved in the AAA pathogenesis, as *PDGFA* is a strong stimulator of this process [38,39]. A similar result for *PDGFA* was obtained in previous studies using a membrane-based complementary DNA expression array and immunohistochemical staining of tissues, in which *PDGFA* was significantly upregulated in AAA compared to the control samples [40,41]; however, the other study showed lower expression of *PDGFA* in aneurysm vs. normal aorta [42]. This discrepancy could result from a type of aneurysm and the site where tissue sample was collected, since, previously, a higher expression of *PDGFA*, together with *PDGFB*, was detected in small vessels of the aneurysmal walls of atherosclerotic AAA, but in inflammatory AAA this expression was significantly lower [43]. Our results confirm that altered *PDGFA* signaling could be involved in AAA development, and management of the *PDGFA* signaling pathway could exert a therapeutic effect, as also previously suggested [44].

Regarding the factors of the TGF family, *TGFB1*, but not *TGFA*, had different expression in the PBMC samples of AAA patients compared to controls (Table 2, Figure 1); however, protein levels of both factors were higher in plasma of the subjects with AAA than in controls (Table 3, Figure 2). In the research literature, studies focused on the role of TGF-alpha in AAA are limited, whereas TGF-beta 1 signaling has been intensively studied on this topic. Some hypotheses about the role of TGF-beta 1 in the development of aneurysm have been proposed; however, there are some controversies [45].

In animal models, TGF-beta 1 was primarily found to protect against AAA by multiple mechanisms. This factor inhibits aortic inflammatory cell infiltration, extracellular matrix degradation, and vascular smooth muscle cells apoptosis, as well as promotes elastin and collagen formation [46,47]. However, these effects were assigned to the canonical, SMAD-dependent signaling pathway [48,49], but activation of other, SMAD-independent TGF-beta 1 pathways (e.g., ERK-dependent pathway) caused exacerbation of the aneurysm and premature death in Marfan syndrome mouse models [50]. In the other study, inhibition of the noncanonical, TGF-beta 1-mediated PI3K/AKT/ID2 signaling pathway ameliorates abdominal aortic aneurysm [51]. From the results of previous studies it could be concluded that the canonical and noncanonical TGF-beta 1 pathways are in dynamic equilibrium in normal aorta. When the TGF-beta 1 pathway is overactivated or suppressed, the noncanonical pathways are excessively activated, causing aortic wall dysfunction and aortic aneurysm [52]. In our study, the expression of *TGFB1* in PBMC was lower in AAA patients compared to controls (Table 2, Figure 1), which presumably may reflect decreased TGF-beta 1 signaling in aortic tissue, previously associated with the development of aneurysm.

In relation to plasma levels of TGF-beta 1 in AAA, previous results are in concordance with our results showing higher plasma levels of this protein in patients with this disease (Table 3, Figure 2). Elevated plasma concentrations of TGF-beta 1 were previously demonstrated in patients with dilatative pathology of ascending aorta [53,54]. In studies on Marfan syndrome models, the cause of higher circulatory levels of TGF-beta 1 in aneurysm was proposed. In such models, the binding capacity of TGF-beta 1 to fibrillin 1 was found to be reduced due to mutations in the FBN1 gene encoding fibrillin 1. This could be a reason for the increase in circulatory levels of TGF-beta 1, which was shown to have prognostic and therapeutic value in this syndrome [55,56,57]. Furthermore, higher plasma levels of TGF-beta 1 could also be an effect of altered binding ability to its receptors, TGFBR1 and TGFBR2, caused by the occurrence of genetic variants in these receptors that were previously associated with aneurysm development [30]. This hypothesis should be validated in future studies.

Although studies on the role of TGF-alpha in AAA are insufficient, this factor has been studied more extensively in cancer, where TGF-alpha was shown to be elevated and associated with increased migration, invasion, and proliferation of cancer cells, as well as enhanced MMP activity and inflammation [58,59,60]. Probably, similar processes could be exerted by TGF-alpha in the aortic wall during AAA development; however, further studies are needed to validate this supposition. Interestingly, TGF-alpha and TGF-beta 1 modulate both the PI3K/Akt and NFκB signaling pathways, which were shown to contribute to aneurysm development. To conclude, higher plasma levels of TGF-alpha and TGF-beta 1 could reflect the involvement of TGF-related signaling pathways (such as the PI3K/Akt and NFκB pathways) in aneurysm development. However, this conclusion needs to be validated in further studies, which should focus not only on TGF-family factors, but also on their effectors that belong to various TGF-related downstream pathways. The systemic application of inhibitors or neutralizing antibodies against these factors in animal models would be a promising approach to make the role of TGF signaling in the aneurysm development more clear.

Studied VEGF-family factors have different expression patterns in PBMC and plasma in the studied groups. The expression of *VEGFA* in PBMC was similar in the AAA and control groups, but plasma VEGF-A levels were higher in AAA patients. In contrast, the expression of *VEGFC* was higher in the AAA group, while plasma VEGF-C levels were lower in this group when compared to the control group. In turn, the expression of *VEGFB* in PBMC was lower in the AAA group vs. control group (Table 2 and Table 3, Figure 1 and Figure 2).

In previous studies, VEGF-A was shown to be overexpressed in the aortic wall of human and experimental AAA and play a crucial role in the development of AAA by regulation of such processes as neoangiogenesis, infiltration of inflammatory cells, MMP activity, and ECM degradation. In animal models of AAA, administration of exogenous VEGF-A was previously found to augment AAA formation, while the reduction of VEGF-A signaling by soluble receptors or inhibitors (e.g., kallistatin) caused inhibition of AAA development [61,62,63,64]. The results obtained in the presented study suggest that the previously reported increased signaling of VEGF-A in AAA tissues is probably reflected in higher plasma levels of VEGF-A, but not in altered expression of *VEGFA* in PBMCs. This conclusion could be supported by other studies, in which elevated plasma levels of VEGF-A were found in aneurysmal diseases [64,65,66,67].

Regarding *VEGFC*, higher expression levels of this gene were found in PBMC of the AAA group compared to the control group (Table 2, Figure 1). A similar effect was also shown in previous studies in abdominal aortic aneurysm wall vs. normal aorta [16,68]. It suggests that enhanced VEGF-C signaling is associated with aneurysm formation and is reflected in circulatory cells. Furthermore, it could also indicate not only enhanced angiogenesis but also lymphangiogenesis, since VEGF-C is a strong stimulator of lymphatic vessels formation [69,70,71].

None of the analyzed regulators of inflammation and angiogenesis was significantly associated with the aneurysm diameter. This observation suggests that the analyzed factors are rather stable and do not change significantly during the disease progression; however, due to the limited number of subjects, further studies are needed to validate this conclusion.

It is obvious that circulatory factors cannot fully reflect the pathological mechanisms ongoing in the vascular wall during AAA development. However, some of them could be secreted from the lesion site and could be reliable indicators of the disease. Furthermore, circulation cells could acquire new features by interacting with pathologically changed aortic sites and could also be used to detect such changes. The results obtained in our study could support this statement, because the altered expressions of some factors in PBMC and plasma were similar to those previously reported in aneurysmal aortic wall. In particular, increased signaling of IL-8, PDGF-A, TGF-beta 1, VEGF-A, and VEGF-C was reported in aneurysm samples in a previous studies; therefore, the changes in the circulatory levels of these factors shown in our study could probably reflect their increased signaling in aneurysm tissue. Future studies should be conducted using both circulatory and site aortic material to confirm this statement.

The presented study has several limitations. In this work, only the main factors were analyzed, and no analysis of dysregulations of receptors or downstream effectors was performed. Such investigations should be carried out in the future to gain deeper insight into the associated molecular pathways that contribute to aneurysm formation. Especially, for some of the studied factors, different receptors and alternative downstream pathways exist and have different biological effects in the context of the studied disease. It remains unclear whether the alterations of proposed biomarkers were predictive of, or a secondary consequence of, the AAA development. Furthermore, the presented work focused on the positive regulators of inflammation and angiogenesis, but the analysis regarding inhibitors was also not carried out. Due to limited resources, plasma protein levels were not measured for all factors used for gene expression analysis. The number of subjects included in the study is limited and statistically significant differences in risk factors and blood parameters between the compared groups are present (Table 1); therefore, the obtained results should be validated in studies with larger cohorts. Finally, the conclusions and hypotheses stated in the discussion section need to be further investigated in future studies.

As shown in our study, genes and proteins with altered expression levels in AAA, after confirmation in extended validation studies, could be used as a diagnostic and therapeutic targets. Analysis of such factors could potentially be used in screening programs for AAA detection, which could result in an increase in the rate of disease detection and a decrease in mortality due to aneurysm rupture. Furthermore, modulation of dysregulated inflammation or angiogenesis pathways could exert a therapeutic effect and expand treatment methods, now limited to invasive procedures such as open surgery and endovascular repair.

## 4. Materials and Methods

### 4.1. Study Participants

The study was performed in accordance with the Declaration of Helsinki. The study procedure was approved by the Bioethics Committee at Medical University of Lublin (decision No. KE-0254/148/2021). The study population included two groups of subjects: 40 patients with AAA (AAA group) and 24 healthy volunteers (control group). The subjects’ qualification was carried out in the Independent Public Clinical Hospital No. 1 in Lublin by a vascular surgeon. Informed and signed consent was obtained from all study subjects.

The AAA group consisted of patients who underwent preoperative aneurysm surveillance (duplex ultrasonography and contrast enhanced spiral computed tomography with volume-rendered reconstructions). Patients were diagnosed with an abdominal aortic aneurysm, diameter ranging from 5.5 to 7.6 cm (mean diameter = 6.3 cm).

Exclusion criteria were established as the following: presence of an inflammatory aneurysm, false aneurysm, thoracic aorta aneurysm, isolated popliteal or iliac artery aneurysm, aortic and/or arterial dissection, stroke, transient ischemic attack, myocardial infarction, type I diabetes mellitus, symptomatic peripheral arterial disease (ABI < 0.8), connective tissue disorders, including rheumatoid disease, impaired hepatic or renal function, corticoid therapy, infection within previous 6 weeks, recent deep venous thrombosis (less than 1 year), pulmonary embolism, inflammatory and/or infectious disease, and cancer.

The control group included healthy, nonsmoking volunteers with excluded blood flow disturbances, abdominal aorta dilatation, and atherosclerosis during physical examination and duplex ultrasound scanning.

The demographic and clinical characteristics of the studied subjects are presented in Table 1.

### 4.2. qPCR Experiments

The expression levels of 10 angiogenesis-related genes (*ANGPT1*, *ANGPT2*, *FGF2*, *PDGFA*, *PDGFB*, *TGFA*, *TGFB1*, *VEGFA*, *VEGFB*, and *VEGFC*) and 8 inflammation-related genes (*CCL2*, *CCL5*, *CSF2*, *CXCL8*, *IL1A*, *IL1B*, *IL6*, and *TNF*) were analyzed in all studied subjects using the qPCR method. The biological material was peripheral blood mononuclear cells (PBMCs) isolated from venous blood collected from subjects after qualification to the study.

Immediately after collection, venous blood was transferred to plasma separation followed by PBMC isolation using gradient centrifugation with Gradisol L reagent (Aqua-Med, Łódź, Poland) according to the standard procedure (previously described in [72]). The collected plasma was aliquoted and stored at −80 °C for ELISA experiments, while the isolated PBMCs were subjected to total RNA extraction using TRI Reagent Solution (Ambion, Austin, TX, USA), according to the manufacturer’s procedure. The quality and quality assessment of the total RNA samples was performed using the NanoDrop ND-1000 spectrophotometer (Thermo Fisher Scientific, Waltham, MA, USA) and the Agilent 2100 Bioanalyzer with the Agilent RNA 6000 Pico Kit (Agilent Technologies, Santa Clara, CA, USA). Total RNA samples with 260/280 ratio higher than 1.8 and with RNA integrity number greater than 7 were subjected to further experiments.

In the next step, total RNA samples were diluted to 100 ng/µL and subsequently reverse transcribed to obtain cDNA using the High Capacity cDNA Reverse Transcription Kit (Applied Biosystems, Foster City, CA, USA) according to the manufacturer’s protocol. The qPCR reactions were prepared in 96-well plates and included 2 µL of cDNA, 10 µL of TaqMan Gene Expression Master Mix (Applied Biosystems, Foster City, CA, USA), 7 µL of nuclease-free water, and 1 µL of TaqMan Gene Expression Assay (Applied Biosystems, Foster City, CA, USA) specific to target genes. The list of used assays is provided in Table 5. *GAPDH* was used as an endogenous control. The qPCR reactions were carried out in triplicate for each sample. Blank reactions (without cDNA) were performed to detect potential contamination by foreign DNA.

The amplification of target genes was carried out using the 7900HT Real-Time Fast System (Applied Biosystems, Foster City, CA, USA). The amplification protocol included initial denaturation (95 °C for 10 min) and 40 amplification cycles (95 °C for 15 s and 60 °C for 1 min in each cycle).

The raw expression data were imported to ExpressionSuite v1.3 software (Life Technologies Corporation, Carlsbad, CA, USA), where the expression levels of analyzed genes were determined as Ct values, i.e., the number of amplification cycle achieved at the intersection between an amplification curve and a threshold line defining the linear phase of the signal growth.

Further steps of the analysis were performed using R 4.3.0 programming software (https://www.r-project.org). To retain only reliable Ct data obtained from high-quality-shaped amplification curves, the Ct values higher than 35 and with the parameter AMPSCORE < 1 were filtered out. Moreover, Ct values flagged as “undetermined” or “inconclusive” were also excluded from the analysis. The amounts of data filtered and retained across the analyzed genes and samples are presented in Appendix A and S4, respectively. The amount of filtered data was similar across samples; therefore, no samples were removed in this step. Regarding filtering across genes, those with a large amount of filtered data (in more than half of subjects in at least one of the study groups) were excluded from the study. These genes were *ANGPT2*, *CSF2*, *IL1A*, and *IL6*. There may be several reasons for the low quality of the results obtained for the excluded genes. The most obvious reason could be very low levels of these genes expression in the used samples, but the presence of inhibitors of the PCR reaction and other issues cannot be excluded. Data regarding the remaining genes were subjected to further steps of analysis. The dataset that resulted after the filtering step contained missing values (4% of the data), which were imputed using multivariate linear regression using Ct values of genes with complete data.

After imputation, the Ct dataset was further analyzed using the ddCt method for relative quantification [73,74]. The Ct values of target genes were normalized using the Ct values of endogenous control (*GAPDH*) by calculating dCt for each sample according to the following formula:dCt = mean Ct of target gene replicates − mean Ct of endogenous control replicates

The uniformity of the dCt data was confirmed on boxplots presenting the distribution of the dCt values in each sample (Appendix A) as well as using PCA analysis (Appendix A).

Differences in expression of target genes between the AAA and control groups were determined by calculating the ddCt values using the following formula:ddCt = mean dCt of target gene in AAA group − mean dCt of target gene in control group

To make ddCt values easier to interpret, the RQ values (RQ = 2^−ddCt^) were calculated for each gene. RQ > 1 means increased gene expression in the group of interest compared to the reference group, while RQ < 1 means decreased gene expression in the group of interest compared to the reference group. RQ = 1 indicates that is no difference in gene expression between the compared groups.

### 4.3. ELISA Experiments

The enzyme-linked immunosorbent assay (ELISA) method was used to measure the concentrations of ANGPT-1, ANGPT-2, TGF-alpha, TGF-beta 1, VEGF-A, and VEGF-C proteins in plasma collected from 40 patients with AAA (AAA group) and 20 healthy volunteers (control group). Commercially available ELISA kits purchased from Biorbyt (Cambridge, UK) were used for this purpose (Table 6).

According to good laboratory practice, the thawed aliquots of plasma samples were centrifuged (2000× *g* for 10 min at 4 °C) to remove platelets and residual cells that could occur in the samples. Prior to the analysis, the concentrations of target proteins in plasma were primarily estimated on the basis of the literature data to select a proper dilution for samples. Such a procedure ensures that the protein concentration obtained in the analysis will fall near the middle of the range in the standard curve. If necessary, samples were diluted using a diluent buffer provided with the ELISA kit.

All ELISA experiments were performed according to the manufacturer’s instructions. Standards of the analyzed proteins were prepared for each experiment using reagents provided with the kits. Eight standard concentrations were performed to draw the standard curve. In addition to standard and samples wells, blank wells that contained a dilution buffer were also used for each experiment.

During the experimental procedure, the ELISA plates were incubated using the DTS-2 incubator (ELMI, Riga, Latvia) under the conditions indicated in the protocol. Plates were read at the appropriate wavelength using the Synergy H1 microplate reader (BioTek, Winooski, VT, USA).

Obtained raw data were analyzed using Gen5 version 3.10 software (BioTek, Winooski, VT, USA). Background absorbance values obtained from the blank wells were subtracted from the values obtained from other wells. The standard curve was plotted as the relative absorbance value of each standard vs. the respective concentration of the standard solution. The protein concentrations in the samples were interpolated from the standard curve. If the samples were diluted, the concentrations obtained from interpolation were multiplied by the dilution factor to obtain the final concentration.

The numbers of samples with obtained concentrations and those where concentrations were too low to be detected are presented in Appendix A. Data consistency was assessed in the PCA plot (Appendix A) to identify potential outlier values. Differences in plasma protein levels between the AAA and control groups were analyzed using the appropriate statistical tests (see the next paragraph).

### 4.4. Statistical Analysis

The statistical analysis was performed in the R 4.3.0 programming environment (https://www.r-project.org). Statistical tests and methods were chosen depending on the type and distribution of the analyzed variables.

Relationships between continuous variables were investigated using correlation and linear regression methods. Correlation analysis was performed using the Spearman rank correlation test implemented in the Hmisc 5.1-0 package (https://cran.r-project.org/web/packages/Hmisc/index.html). Univariate and multivariate linear regression models were constructed using the lm base function in R.

The associations between continuous and categorical variables were analyzed using the logistic regression as well as the two-sided Student’s *t*-test (t.test function in R) or the two-sided Mann–Whitney U-test (wilcox.test function in R), depending on the normality of the data assessed using the Shapiro–Wilk test (shapiro.test function in R). If the distributions of the analyzed variables in both compared groups were defined as normal (*p* > 0.05 in the Shapiro–Wilk test), the parametric Student’s *t*-test was used, while if the distributions of the dCt values in at least one of the compared groups were defined as not normal (*p* < 0.05 in the Shapiro–Wilk test), the nonparametric Mann–Whitney U-test was used. Univariate and multivariate logistic regression was performed using the glm() base function in R.

The relationships between categorical variables were analyzed using the Fisher’s exact test (the fisher.test function in R).

ROC analysis and plots were performed using the pROC package 1.18.0 [75] (https://cran.r-project.org/web/packages/pROC/index.html).

Results obtained with *p* < 0.05 were considered statistically significant. Visualizations were generated using the ggplot2 3.4.2 package (https://ggplot2.tidyverse.org/) in R unless otherwise noted.

## 5. Conclusions

Demonstrated dysregulations in key regulators of angiogenesis and inflammation in patients with AAA indicate that the abnormal course of these processes contributes to AAA formation. Angiogenesis pathways associated with ANGPT-1, PDGF-A, TGF-beta 1, and VEGF signaling, as well as IL-8 inflammation pathways, are probably involved in the development of this disease. The presented study provides promising diagnostic or therapeutic targets; however, further studies are needed to validate the obtained results.

## Figures and Tables

**Figure 1 ijms-24-12087-f001:**
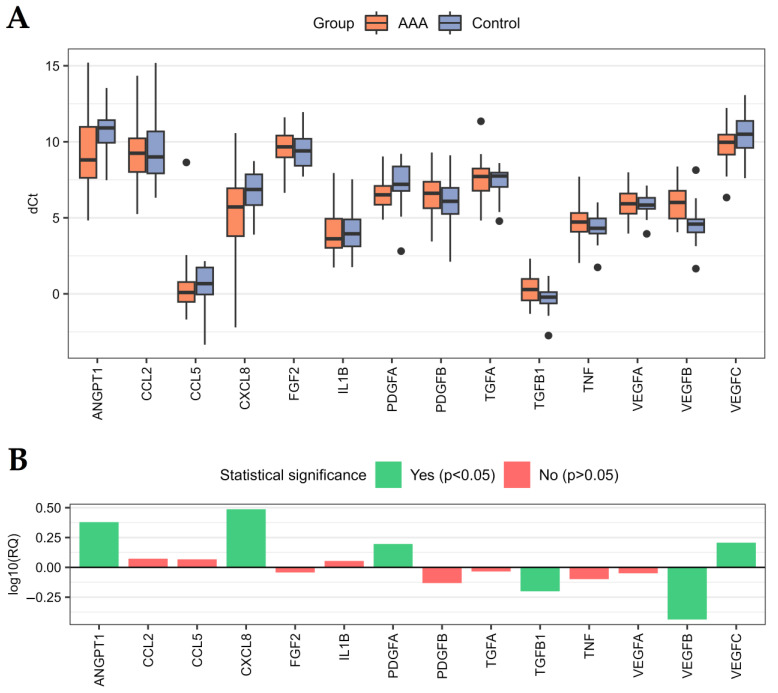
Expression of 14 analyzed genes in the AAA and control groups. (**A**) Distribution of normalized expression (dCt values) of analyzed genes in the studied groups. Whiskers reach the most distant point in the doubled interquartile range (samples located outside the whiskers are presented as round points), boxes range between 25% and 75% quartile, and horizontal lines inside boxes mark the median value. For proper interpretation of the plot, the inverse relationship between the dCt values and the expression levels should be taken into account (refer to panel (**B**)). (**B**) RQ values obtained for each analyzed gene in comparison between the AAA and control groups. AAA—group of patients with abdominal aortic aneurysm, Control—group of non-AAA subjects.

**Figure 2 ijms-24-12087-f002:**
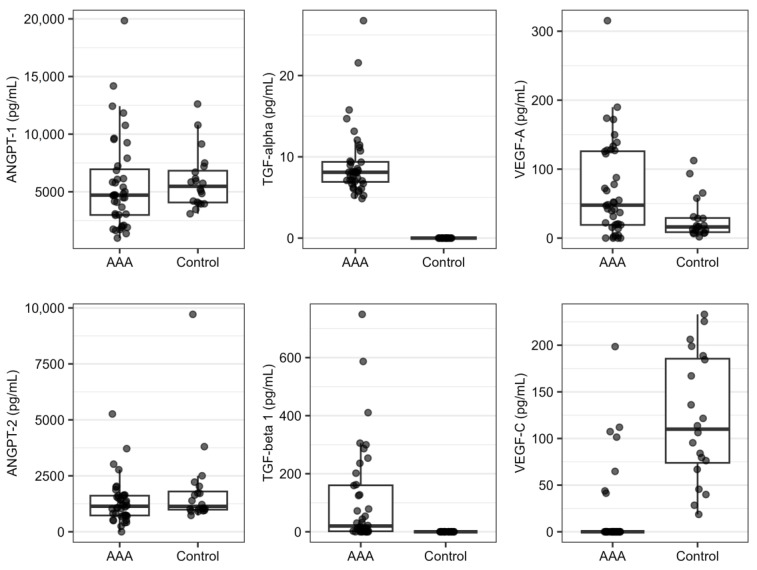
Distributions of plasma levels obtained for analyzed proteins in patients with AAA (AAA group) and healthy controls (control group). Whiskers reach the most distant point in the 1.5 interquartile range, boxes range between 25% and 75% quartile, and horizontal lines inside boxes mark the median value.

**Figure 3 ijms-24-12087-f003:**
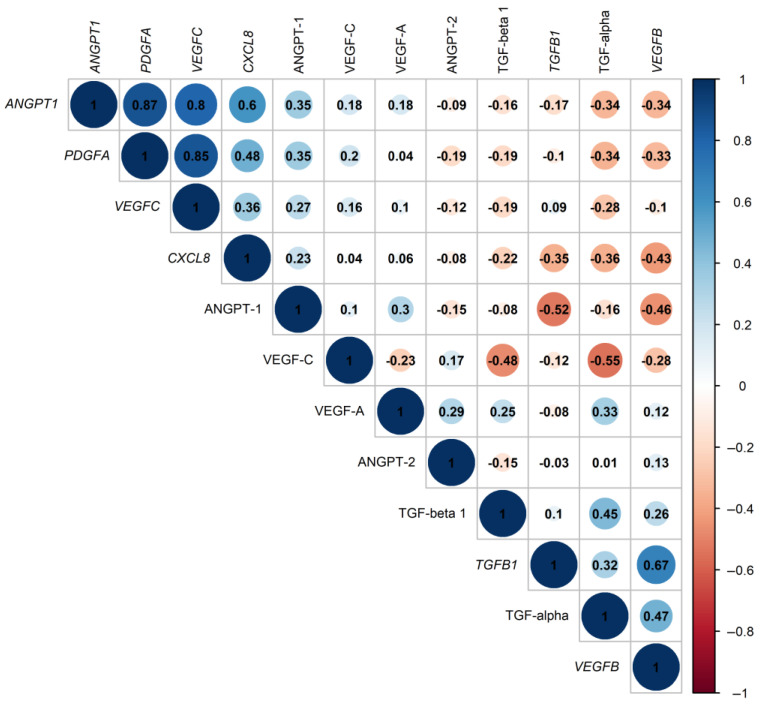
Correlation coefficients obtained for normalized expression levels of 6 selected genes (*ANGPT1*, *CXCL8*, *PDGFA*, *TGFB1*, *VEGFB*, and *VEGFC*) and plasma levels of 6 proteins (ANGPT-1, ANGPT-2, TGF-alpha, TGF-beta 1, VEGF-A, and VEGF-C) analyzed in the AAA and control groups, calculated using Spearman rank correlation test. The plot was generated using the corrplot 0.92 package in R.

**Table 1 ijms-24-12087-t001:** Demographical and clinical data of study subjects.

Characteristic	AAA Group (n = 40)	Control Group (n = 24)	*p* ^1^
Age	59.2 ± 9.47 (45–80)	55.6 ± 9.13 (35–73)	n.s.
Sex male/female	34 (85.0%)/6 (15.0%)	14 (58.3%)/10 (41.7%)	3.455 × 10^−2^
Body mass index (BMI)	26.8 ± 4.27 (19.5–35.1)	25.2 ± 3.09 (21.2–32.9)	n.s.
Smoking	15 (37.5%)	0 (0%)	4.324 × 10^−4^
Hypertension	7 (17.5%)	4 (16.7%)	n.s.
Hypercholesterolemia	17 (42.5%)	7 (29.2%)	n.s.
Hypertriglyceridemia	3 (7.5%)	0 (0%)	n.s.
LDL (mg/dL)	109 ± 14.2 (79–151)	102 ± 9.9 (84–117)	1.853 × 10^−2^
HDL (mg/dL)	40.7 ± 3.70 (31–46)	41.2 ± 2.88 (35–47)	n.s.
Cholesterol (mg/dL)	206 ± 22.3 (143–302)	191 ± 9.14 (178–204)	1.720 × 10^−5^
Creatinine (mg/dL)	0.80 ± 0.16 (0.38–1.08)	0.8 ± 0.13 (0.45–1.03)	n.s.
Urea (mg/dL)	35.5 ± 5.7 (25–44)	36.3 ± 2.57 (31–41)	n.s.
C-reactive protein (mg/L)	4.3 ± 1.3 (1.1–6.8)	2.5 ± 0.9 (1.1–4.3)	3.226 × 10^−8^
Fibrinogen (mg/dL)	177 ± 45.4 (121–316)	195 ± 40.7 (112–278)	2.371 × 10^−2^
Homocysteine (µmol/L)	8.0 ± 2.0 (3.6–13.8)	7.2 ± 1.3 (5.1–10.8)	n.s.

^1^ Statistical significance of differences between the group of patients with abdominal aortic aneurysm (AAA group) and the group of control subjects (control group) was calculated for continuous-type variables (age, BMI, and biochemical blood parameters) using the two-sided Student’s *t*-test or the two-sided Mann–Whitney U-test (depending on the normality of the data). For categorical-type variables (sex, smoking, hypertension, hypercholesterolemia, and hypertriglyceridemia), the two-sided Fisher’s exact test was used. Continuous-type variables are presented as mean ± SD, and range in brackets. Categorical-type variables are presented as count and percentage in brackets (n.s.—not significant (*p* > 0.05)).

**Table 2 ijms-24-12087-t002:** Genes differentially expressed between AAA and control groups with statistical significance (*p* < 0.05).

Gene Symbol	Gene Name	Relative Quantification		ROC	Univariate Logistic Regression	
		RQ	*p*	ROC-AUC	OR	*p*
*ANGPT1*	Angiopoietin 1	2.395	1.19 × 10^−2^	0.689	1.328	3.11 × 10^−2^
*CXCL8*	C–X–C motif chemokine ligand 8	3.069	1.36 × 10^−2^	0.684	1.376	2.29 × 10^−2^
*PDGFA*	Platelet-derived growth factor subunit A	1.571	1.03 × 10^−2^	0.693	1.577	4.89 × 10^−2^
*TGFB1*	Transforming growth factor beta 1	0.630	4.45 × 10^−3^	0.688	0.423	1.10 × 10^−2^
*VEGFB*	Vascular endothelial growth factor B	0.365	1.79 × 10^−5^	0.809	0.333	7.58 × 10^−4^
*VEGFC*	Vascular endothelial growth factor C	1.610	4.37 × 10^−2^	0.645	1.628	4.03 × 10^−2^

Provided gene symbols and gene names are in accordance with actual nomenclature in HUGO Gene Nomenclature Committee (HGNC) (https://www.genenames.org/, accessed on 1 June 2023). OR—odds ratio, ROC-AUC—area under receiver operating characteristics curve, RQ—relative quantity (fold change calculated by formula: RQ = 2^−ddCt^).

**Table 3 ijms-24-12087-t003:** Differences in analyzed plasma proteins levels between AAA and control group.

Protein Symbol	Protein Name	Mean Concentration (pg/mL)		*p*	AUC-ROC
		AAA	Control		
ANGPT-1	Angiopoietin-1	5643.89 ± 3994.20	5971.96 ± 2490.58	2.58 × 10^−1^	0.591
ANGPT-2	Angiopoietin-2	1330.80 ± 981.72	1875.85 ± 1986.56	2.01 × 10^−1^	0.398
TGF-alpha	Protransforming growth factor alpha	9.12 ± 4.31	0.00 ± 0.00	1.73 × 10^−10^	1.000
TGF-beta 1	Transforming growth factor beta-1 proprotein	108.38 ± 170.42	0.00 ± 0.00	1.25 × 10^−7^	0.900
VEGF-A	Vascular endothelial growth factor A	69.89 ± 68.97	28.03 ± 30.56	1.29 × 10^−2^	0.699
VEGF-C	Vascular endothelial growth factor C	16.73 ± 42.49	120.82 ± 68.07	3.29 × 10^−9^	0.931

Provided protein names are in accordance with UniProt database (release 2023_02, https://www.uniprot.org/, accessed on 1 June 2023). *p*—statistical significance calculated by two-sided Mann–Whitney U-test; ROC-AUC—area under receiver operating characteristics curve.

**Table 4 ijms-24-12087-t004:** Statistically significant correlations (*p* < 0.05) between genes or proteins and blood biochemical parameters.

Correlated Variables	Correlation		Univariate Linear Regression	
	R	*p*	β	*p*
*VEGFB*—fibrynogen	−0.34	6.19 × 10^−3^	9.51 × 10^−3^	1.71 × 10^−2^
*TGFB1*—fibrynogen	−0.27	2.96 × 10^−2^	6.69 × 10^−3^	1.29 × 10^−2^
*VEGFB*—homocysteine	0.29	2.15 × 10^−2^	1.96 × 10^−1^	4.40 × 10^−2^
TGF-alpha—CRP	0.53	1.60 × 10^−5^	1.931	5.50 × 10^−5^
TGF-beta 1—cholesterol	0.51	3.49 × 10^−5^	3.864	1.68 × 10^−5^
VEGF-C—CRP	−0.51	3.80 × 10^−5^	−20.452	1.18 × 10^−3^
TGF-alpha—cholesterol	0.35	5.66 × 10^−3^	9.92 × 10^−2^	5.09 × 10^−3^
VEGF-C—LDL	−0.35	6.31 × 10^−3^	−1.606	2.00 × 10^−2^
VEGF-A—LDL	0.34	7.52 × 10^−3^	1.612	6.50 × 10^−3^
VEGF-A—CRP	0.29	2.61 × 10^−2^	11.119	4.74 × 10^−2^

R—correlation coefficient, β—regression coefficient.

**Table 5 ijms-24-12087-t005:** TaqMan Gene Expression Assays used for the study.

Assay ID	Gene Symbol	Gene Name	Amplicon Length
Hs00919201_m1	*ANGPT1*	Angiopoietin 1	119
Hs00169867_m1	*ANGPT2*	Angiopoietin 2	73
Hs00234140_m1	*CCL2*	C–C motif chemokine ligand 2	101
Hs99999048_m1	*CCL5*	C–C motif chemokine ligand 5	98
Hs00929873_m1	*CSF2*	Colony-stimulating factor 2	85
Hs00174103_m1	*CXCL8*	C–X–C motif chemokine ligand 8	101
Hs99999905_m1	*GAPDH*	Glyceraldehyde-3-phosphate dehydrogenase	122
Hs00174092_m1	*IL1A*	Interleukin 1 alpha	69
Hs01555410_m1	*IL1B*	Interleukin 1 beta	91
Hs00174131_m1	*IL6*	Interleukin 6	95
Hs00266645_m1	*FGF2*	Fibroblast growth factor 2	82
Hs00234994_m1	*PDGFA*	Platelet-derived growth factor subunit A	93
Hs00966522_m1	*PDGFB*	Platelet-derived growth factor subunit B	56
Hs00608187_m1	*TGFA*	Transforming growth factor alpha	70
Hs00998133_m1	*TGFB1*	Transforming growth factor beta 1	57
Hs00174128_m1	*TNF*	Tumor necrosis factor	80
Hs00900055_m1	*VEGFA*	Vascular endothelial growth factor A	59
Hs00173634_m1	*VEGFB*	Vascular endothelial growth factor B	69
Hs01099203_m1	*VEGFC*	Vascular endothelial growth factor C	66

Provided gene symbols and gene names are in accordance with actual nomenclature in HUGO Gene Nomenclature Committee (HGNC) (https://www.genenames.org/, accessed on 1 June 2023).

**Table 6 ijms-24-12087-t006:** ELISA kits used for the study.

ELISA Kit ID	Protein Symbol	Protein Name	Quantitative Range (pg/mL)
orb138056	ANGPT-1	Angiopoietin-1	156.25–10,000
orb146693	ANGPT-2	Angiopoietin-2	156.25–10,000
orb50169	TGF-alpha	Protransforming growth factor alpha	15.625–1000
orb50103	TGF-beta 1	Transforming growth factor beta 1 proprotein	15.625–1000
orb50119	VEGF-A	Vascular endothelial growth factor A	31.25–2000
orb50131	VEGF-C	Vascular endothelial growth factor C	62.5–4000

Provided protein names are in accordance with actual nomenclature in Uniprot database (https://www.uniprot.org/, accessed on 1 June 2023).

## Data Availability

The data used for this study are openly available in FigShare repository at https://doi.org/10.6084/m9.figshare.23791485.v1.

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
