# Peer review of "Dysregulations of Key Regulators of Angiogenesis and Inflammation in Abdominal Aortic Aneurysm"

_ijms, 2023, doi:10.3390/ijms241512087_

Round 1
Reviewer 1 Report
This manuscript reports the quantification of proteins that mediate angiogenesis and inflammation in the blood plasma and peripheral blood mononuclear cells (PBMCs) of patients with abdominal aortic aneurysm (AAA). The authors use qPCR approaches to quantify gene expression in PBMCs and ELISAs to detect protein expression in plasma, with the goal of identifying biomarkers that could serve as diagnostic and therapeutic targets. Overall, the results are well documented and clearly presented. The manuscript would be improved if the following points were addressed.
1. What is the meaning of the blue dots in Figure 1A?
2. On page 5, the authors state that “In samples in which the measurements were out of the range of used ELISA kits, the concentrations were calculated by extrapolation of the standard curve.” It is far better to dilute the sample so that the measured value falls on the standard curve. The standard curve may not be linear outside the recommended range.
3. Thrombospondin-1 (TSP-1) is an important inhibitor of angiogenesis that may compensate for the up-regulation of pro-angiogenic factors. The study would be more complete if TSP-1 levels were also quantified.
4. Throughout the manuscript, the authors use an italic font to distinguish genes from proteins. This should be done in Figure 3.
Author Response
Thank you for your kind opinion on our work and suggestions. We have introduced corrections accordingly to your comments.
Reviewer’s comment:
- What is the meaning of the blue dots in Figure 1A?
Authors’ response:
The dots in Figure 1A define samples (observation points) that are located outside the whiskers of the box plot. This information was missing and was added to the legend of Figure 1 and Figure S2. There are only ten such data points and these distant points rather illustrate a biological variability of data and do not introduce a bias to the results, since in performed quality control of the data, a good homogeneity of data was observed and no outlier samples were identified (see Figures S1-S3, and also Figure S8).
Reviewer’s comment:
- On page 5, the authors state that “In samples in which the measurements were out of the range of used ELISA kits, the concentrations were calculated by extrapolation of the standard curve.” It is far better to dilute the sample so that the measured value falls on the standard curve. The standard curve may not be linear outside the recommended range.
Authors’ response:
Thank you for paying our attention to this point; this is an evident misstatement in this sentence. In our experiments, when the concentrations were outside the range of the ELISA kit, the values were not extrapolated from the standard curve, but, according to your comment (with which we fully agree) and as we mentioned in the Material and Methods section (see paragraph 4.2. lines 565-569), the plasma samples were diluted to fall within the standard curve.
We corrected the sentence you mentioned in this comment.
Reviewer’s comment:
- Thrombospondin-1 (TSP-1) is an important inhibitor of angiogenesis that may compensate for the up-regulation of pro-angiogenic factors. The study would be more complete if TSP-1 levels were also quantified.
Authors’ response:
We fully agree with your comment. Information about the influence of factors that inhibit angiogenesis (such as TSP-1, according to your very interesting suggestion) is also important and would make our work more complete. However, due to limited resources and the amount of samples, we are able to analyze a restricted number of genes and proteins associated with the regulation of angiogenesis. Therefore, in our work, we focused on activators of angiogenesis since there are evidences that this process is enhanced in AAA.
You pointed out a main limitation of our study, where only selected, main regulators were analyzed, but regulatory mechanisms of angiogenesis are very complex and involve many factors, which should be included in future investigations. This limitation was addressed at the end of the Discussion section (lines 427-432).
Thank you for your valuable suggestion, which will be considered by our research team during planning our future projects. We decided to publish our present work to share our findings with the scientific community and to indicate directions for further research on this topic for other researchers. Further studies could focus in more detail on the signaling pathways that were shown in our study to probably be involved in AAA development, as well as other pathways associated with activation and inhibition of inflammation or angiogenesis.
Reviewer’s comment:
- Throughout the manuscript, the authors use an italic font to distinguish genes from proteins. This should be done in Figure 3.
Authors’ response:
Figure 3 was corrected according to your comment. Now, Figure 3 is easier to interpret by the readers.
Reviewer 2 Report
The manuscript submitted describes dysregulation of possible key factors of angiogenesis and inflammation related to the pathophysiologic process of abdominal aortic aneurysms.
The authors have collected blood samples and analyzed quantitative PCR expression of markers of angiogenesis and inflammation and determined serum protein levels from patients with abdominal aortic aneurysms compared to healthy controls.
Some of these markers were upregulated, some showed a reduced expression or protein levels and might be associated with the development or progression of abdominal aortic aneurysms. However, the findings remain more speculative and it is not clear if these changes are the cause or just a secondary consequence of the aortic aneurysmal disease.
1. The authors mention that none of the factors investigated were correlated to the aortic aneurysm diameter. This observation seems to indicate that the markers are not involved in regulation of diameter progression.
2. Well it is not a new question; however, please comment about the impact of soluble markers or proteins found in peripheral blood cells how these markers can have an effect on localized changes in the infrarenal aortic wall. These changes could probably work as cofactors, but others pathogenetic changes should be present in the aortic wall.
3. Most interesting are probably the increased levels of TGF alpha and TGF beta 1 in the patients with aortic aneurysms with very low levels or undetectable levels in the healthy control. Are there any experimental studies with knockout animals or si RNA blockade of these genes regarding the development of abdominal aortic aneurysms. Nevertheless, these markers were also suspected to be involved in tumor development and local tumor associated inflammation.
4. Please, summarize or try to confirm that the speculations why the changes in marker expression could be the causative factors and which are probably secondary cell proteins released from local sites of increased degenerative cell metabolism with shedding of these factors into the circulation.
5. What would be next step to make the relevance of these findings more clear ? Do the authors conclude to use for example neutralizing antibodies against TGF alpha or against TGF beta 1 or systemic application of inhibitors in an animal model to support these findings ?
6. At the moment the authors have to mention that they describe an altered PCR based expression in peripheral blood cells and different protein levels of some factors related to angiogenesis and inflammation in patients with abdominal aortic aneurysms compared to healthy controls, although the relevance of these findings so far remain unclear. Or give further arguments to show the impact of these findings for possible diagnostic or therapeutic strategies, as mentioned at the end of the abstract.
7. page 5 line 186/187: please make clear that levels of TGF a and TGF beta 1 were not detectable in the control group instead of ... were not determined..
8. minor: please mention the p values in a stringent way. Do not change from > 0.001 to 10-5 numbers.
9. in table 1: just mention n.s. (not significant) when the difference is not significant.
minor revisions
Author Response
Thank you for your opinion on our work and we appreciate constructive criticism. We introduced the suggested corrections that significantly improved our manuscript.
Reviewer’s comment:
- The authors mention that none of the factors investigated were correlated to the aortic aneurysm diameter. This observation seems to indicate that the markers are not involved in regulation of diameter progression.
Authors’ response:
We agree with your statement. We were hopeful that some of the analyzed genes and proteins could be used to predict the aneurysm diameter, but none of the analyzed factors was significantly correlated with the aneurysm size. It suggests that the pathological processes related to the analyzed regulators of inflammation and angiogenesis are rather stable and do not change during the disease progression. However, due to the limited number of subjects and analyzed factors, further studies are needed to validate this suggestion. This remark was missing from our manuscript and was added to the Discussion section (lines 410-414).
Reviewer’s comment:
- Well it is not a new question; however, please comment about the impact of soluble markers or proteins found in peripheral blood cells how these markers can have an effect on localized changes in the infrarenal aortic wall. These changes could probably work as cofactors, but others pathogenetic changes should be present in the aortic wall.
Authors’ response:
Thank you for pointing out this aspect, too little attention has been paid to this subject in our manuscript.
It is obvious that circulatory factors cannot fully reflect all pathomechanisms ongoing in the vascular wall; however, some proteins could be secreted from the lesion site and be reliable indicators of the disease. Furthermore, circulation cells could acquire new features by interacting with pathologically changing aortic site and released factors, thus also could be used to detect such changes. The results obtained in our study could support this statement, because the altered expression of some factors in PBMC and plasma was similar to those previously reported in the aneurysmal aortic wall (e.g. higher expression of CXCL8, PDGFA, and VEGFC, as well as higher levels of TGF-beta 1 and VEGF-A).
The comment regarding this aspect was added to the Discussion section (lines 415-426).
Reviewer’s comment:
- Most interesting are probably the increased levels of TGF alpha and TGF beta 1 in the patients with aortic aneurysms with very low levels or undetectable levels in the healthy control. Are there any experimental studies with knockout animals or si RNA blockade of these genes regarding the development of abdominal aortic aneurysms. Nevertheless, these markers were also suspected to be involved in tumor development and local tumor associated inflammation.
Authors’ response:
Studies that investigate TGF signaling in animal models of AAA were previously performed and relevant review articles on this topic were published (see references nr 46-50). After the analysis of the literature, it is clear that the TGF-alpha and especially TGF-beta 1 signaling pathways are very complex and the effects of the dysregulations found in our study are difficult to predict.
Studies on the role of TGF-alpha in AAA are limited, but, according to your comment, elevated levels of this factor were found in in cancer and were associated with increased migration, invasion, and proliferation of cancer cells, as well as with enhanced MMP activity and inflammation. These observations are suspected to be an effects of the activation of such downstream pathways as PI3K/Akt and NFκB signaling. Probably, similar processes could be exerted by TGF-alpha in the aortic wall during AAA development; however, more studies are needed to validate this conclusion.
Regarding TGF-beta 1, the impact of this regulator on AAA seems to be different depending on the activated downstream pathway: the canonical pathway has a protective effect, while the non-canonical ones augment disease progression. In our study, the analysis of downstream TGF pathways was not included, and this limitation was addressed in the discussion section (lines 427-432). Interestingly, TGF-alpha and TGF-beta 1 modulate both PI3K/Akt and NFκB signaling pathways, which were shown to contribute to aneurysm development.
These aspects were added to the manuscript (lines 371-385).
Reviewer’s comment:
- Please, summarize or try to confirm that the speculations why the changes in marker expression could be the causative factors and which are probably secondary cell proteins released from local sites of increased degenerative cell metabolism with shedding of these factors into the circulation.
Authors’ response:
As we mentioned in the response to comment No. 2, observed changes in protein plasma levels in AAA correspond to the changes previously found in aortic wall tissue; therefore, it could indicate that these proteins could be released from the site of the aneurysm and be a secondary hallmark of a disease. However, in our study, it was not possible to definitely determine which factors are causative and which are a secondary to the pathology. Such investigations require different design and methodological aspects. This limitation was added to the Discussion section (lines 432-434).
Reviewer’s comment:
- What would be next step to make the relevance of these findings more clear ? Do the authors conclude to use for example neutralizing antibodies against TGF alpha or against TGF beta 1 or systemic application of inhibitors in an animal model to support these findings?
Authors’ response:
In many points of our manuscript, especially in the Discussion section, we proposed further research directions to be taken in the future. We agree with your comment that those studies should include investigations on animal models using inhibitors or neutralizing antibodies against TGF-family factors and their downstream effectors to make the role of TGF signaling in the aneurysm development more clear. Such a statement was added to the Discussion section (lines 382-385).
Reviewer’s comment:
- At the moment the authors have to mention that they describe an altered PCR based expression in peripheral blood cells and different protein levels of some factors related to angiogenesis and inflammation in patients with abdominal aortic aneurysms compared to healthy controls, although the relevance of these findings so far remain unclear. Or give further arguments to show the impact of these findings for possible diagnostic or therapeutic strategies, as mentioned at the end of the abstract.
Authors’ response:
We are aware of the limitations of our work, which could impact the relevance of presented results (see lines 427-442). The number of the analyzed genes and proteins is limited, and the sample size is not sufficient to draw a definitive conclusion about the observed associations. However, the main objective of the presented work was not to finally determine signatures of AAA ready for clinical application, but to propose a set of candidates for biomarkers that could be confirmed in further studies with much larger number of samples. However, despite these limitations, the results obtained give a clear, primarily image of the changes in the expression of main regulators of inflammation and angiogenesis associated with aneurysm. In the discussion section, we draw many conclusions and hypotheses which provide many new research directions for future studies that could be used by other researchers who work on the same topic.
Showed in our study genes and proteins with altered expression levels in AAA, after confirmation in extended validation studies, could be used as a diagnostic and therapeutic targets. Analysis of such factors could potentially be used in screening programs for AAA detection, which could result in an increase in the rate of disease detection and a decrease in mortality due to aneurysm rupture. Furthermore, modulation of dysregulated inflammation or angiogenesis pathways could exert a therapeutic effect and expand treatment methods, now limited to invasive procedures such as open surgery and endovascular repair. This statement was added at the end of the Discussion section (lines 443-450).
Reviewer’s comment:
- page 5 line 186/187: please make clear that levels of TGF a and TGF beta 1 were not detectable in the control group instead of ... were not determined..
Authors’ response:
Thank you for this remark; the word ‘determined’ is ambiguous in this context, the word ‘detected’ sounds much better. The suggested correction was made.
Reviewer’s comments:
- minor: please mention the p values in a stringent way. Do not change from > 0.001 to 10-5numbers.
- in table 1: just mention n.s. (not significant) when the difference is not significant.
Authors’ response:
The p values were unified through the manuscript and suggested corrections were introduced.
Reviewer 3 Report
This article reports on a study investigating the potential role of biomarkers of AAA. Regulators of inflammation and angiogenesis were investigated both in gene expression and plasma protein levels. They found that some regulators of angiogenesis and inflammation were associated with AAA. The article is overall well written and the scientific methodology is sound. The limitations of the study are correctly acknowledged. I have no request for revisions.
Author Response
Thank you for your kind opinion on our work.
Round 2
Reviewer 1 Report
The authors have responded well to the reviewers' comments.
Reviewer 2 Report
The authors have further improved their manuscript, mentioned the possible implication for aortic wall degeneration and also made clear the limitations of the study. Now the paper is acceptable for publication.
English needs minor corrections.